# Effect of Rotation of the Principal Stress Axes Relative to the Material on the Evolution of Material Properties in Severe Plastic Deformation Processes

**DOI:** 10.3390/ma13204667

**Published:** 2020-10-20

**Authors:** Marko Vilotic, Leposava Sidjanin, Sergei Alexandrov, Lihui Lang

**Affiliations:** 1Faculty of Technical Sciences, University of Novi Sad, 21000 Novi Sad, Serbia; markovil@uns.ac.rs (M.V.); lepas@uns.ac.rs (L.S.); 2School of Mechanical Engineering and Automation, Beihang University, Beijing 100191, China; sergei_alexandrov@spartak.ru; 3Ishlinsky Institute for Problems in Mechanics RAS, 119526 Moscow, Russia; 4Federal State Autonomous Educational Institution of Higher Education, South Ural State University (National Research University), 454080 Chelyabinsk, Russia

**Keywords:** severe plastic deformation, V-shape dies, upsetting, ideal flow, microstructure, microhardness

## Abstract

Severe plastic deformation (SPD) processes are widely used for improving material properties. A distinguishing feature of many SPD processes is that the principal axes of the stress tensor intensively rotate relative to the material. Nevertheless, no measure of this rotation is involved in the constitutive equations that predict the evolution of material properties. In particular, a typical way of describing the effect of SPD processes on material properties is to show the dependence of various parameters that characterize these properties on the equivalent strain. However, the same level of the equivalent strain can be achieved in a process in which the principal axes of the stress tensor do not rotate relative to the material. It is, therefore, vital to understand which properties are dependent and which properties are independent of the rotation of the principal axes of the stress tensor relative to the material. In the present paper, a new multistage SPD process is designed such that the principal stress axes do not rotate relative to the material during each stage of the process but the directions of the major and minor principal stresses interchange between two subsequent stages. The process is practically plane strain, and it may be named the process of upsetting by V-shape dies. In addition, axisymmetric compression by Rastegaev’s method is conducted. In this case, the principal stress axes are fixed in the material throughout the entire process of deformation. Material properties and microstructure generated in the two processes above are compared to reveal the effect of the rotation of the principal stress axes relative to the material on the evolution of these properties.

## 1. Introduction

Severe plastic deformation (SPD) processes are attracting an increasing interest due to material properties generated by these processes [1]. In contrast to conventional metal forming processes, many SPD processes do not practically change the shape of the original specimen. The most widely used processes of this type are equal-channel angular pressing (ECAP) (for example, [2,3,4,5]) and high-pressure torsion (HPT) (for example, [6,7,8,9,10]). The distribution of material properties is quite uniform in specimens produced by such processes because the state of stress and strain is quite uniform in the plastic region. Other SPD processes are somehow similar to conventional metal forming processes (for example, [11,12,13]). Those are multistage processes in which considerable plastic strain is accumulated by increasing the number of stages. The state of stress should be compressive enough to prevent the initiation of fracture. The distribution of stress and strain in the plastic region is not uniform. Therefore, the distribution of material properties in the final product can be uniform only if the change in these properties is negligible once the equivalent strain has attained a certain level. Special SPD methods have been developed for sheet metals, for example [14].

The effect of SPD processing on material behavior is the subject of many publications, for example, [15]. Independently of a specific SPD process, its primary goal is to improve material properties. Numerous parameters are adopted to quantify these properties. Typical parameters are the grain size and hardness (for example, [16,17,18,19,20,21,22]). In these works, the parameters are presented as functions of the equivalent strain. However, the same level of the equivalent strain can be achieved in processes that are usually not classified as SPD processes. For example, multistage extrusion through the sigmoidal dies proposed in [23,24] can be used for producing quite large strain without fracture. Moreover, the distribution of hardness in the strip after plane strain drawing through the sigmoidal die is quite uniform [25]. The processes described in [23,24] are ideal flow processes [26]. An essential difference between the ideal flow processes and SPD processes, such as equal-channel angular pressing and high pressure torsion, is that the trajectories of the major principal stress are material lines in the former processes and the principal stress axes intensively rotate relative to the material in the latter processes. It is not clear whether or not this rotation has an essential effect on material properties. A general theoretical method to reveal this possible effect has been proposed in [27]. The present paper attempts to apply this method experimentally.

The SPD process proposed is multistage upsetting between two V-shape dies. The initial cross-section of specimens is a square. The cross-section of specimens after each stage of deformation is a rhombus. It is evident that the process has two axes of symmetry. The evolution of material properties and microstructure is studied on these axes only. During each stage of the process, the ideal flow conditions are satisfied for all points on the axes of symmetry. However, the directions of the major and minor principal stresses interchange relative to the material between two subsequent stages. Because of this change in the directions of the principal stresses, this process can be classified as a special SPD process in which the principal axes of the stress tensor rotate relative to the material in a jump-like manner.

One of the most straightforward ideal flow processes is the compression of a cylinder using Rastegaev’s method [28]. An additional advantage of this process for the present research is that it is relatively easy to control the equivalent strain in the specimen by the displacement of the tool. Having the value of the equivalent strain at which the material properties have been quantified in the SPD process, it is possible to achieve the same strain and then measure these properties in the cylinder. This approach is used in the present paper to show the effect of rotation of the principal stress axes relative to the material on hardness and microstructure.

## 2. General Approach

A general approach for developing a new type of continuum mechanics constitutive equations for predicting the evolution of material properties in SPD processes has been proposed in [27]. For the reason of readability, this approach is outlined in the present section.

As an example, consider the evolution of hardness or microhardness in the course of SPD processes. It is usually assumed that these properties are controlled by the equivalent strain, ε_eq_, or a similar quantity [11,16,21,29,30]. However, it has been shown in [31,32] that the strain reversal affects the evolution of material properties in HPT processing. These results already show that the equivalent strain alone cannot adequately describe the evolution of material properties in the course of SPD processes. It has been noted in [27] that the rotation of principal stress axes relative to the material is usually significant in SPD processes. It is then reasonable to assume that a measure of this rotation, Ω, may affect the evolution of material properties. Figure 1 schematically shows several paths in the ε_eq_—Ω space. The equivalent strain attains a value of ε_0_ at the end of each path.

If Ω has no effect on the evolution of material properties then these properties should be identical at points A_0_, A_1_, A_2_, and A_3_. Point A_0_ corresponds to an ideal flow path in which it is possible to put Ω = 0 without a loss of generality. On the other hand, essential differences in the material properties suggest that Ω should be included in the corresponding constitutive equations. Experimental verification of which hypothesis is correct is feasible. It is necessary to produce the same equivalent strain along several paths, including an ideal flow path. The present paper is devoted to such verification.

## 3. Experimental Procedures

### 3.1. Deformation Processes

Two deformation processes were employed to reveal a possible effect of SPD on material properties. One of these processes was the upsetting of a circular cylinder by Rastegaev’s method [28]. Using this method, one can achieve a high level of homogeneous strain in the cylinder. This process is an ideal flow process [26] in which the principal stress axes do not rotate relative to the material.

The other process can be classified as a new multistage SPD process. Its schematic diagram is shown in Figure 2. The specimen was repeatedly deformed between two identical V-shape dies. The lower die was motionless, and the upper die moved vertically. After each stage of deformation, the specimen was rotated by 90°. This process and the specimen processed have three planes of symmetry. One of these planes (plane *C*) coincides with the central cross-section of the billet throughout the process of deformation (Figure 3). The other two planes are the horizontal and vertical planes of symmetry of the billet throughout the process of deformation. These planes are denoted as plane *H* and plane *V*, respectively.

The intersections of each of these planes with a cross-section parallel to plane *C* generate two axes of symmetry of the cross-section. These axes are denoted as *h* and *v*, respectively. The intersection of planes *H* and *V* generates axis *c*. All planes and axes above are illustrated in Figure 3.

The principal stress directions are known on axes *h* and *v*. Moreover, it is obvious that the major principal stress direction coincides with axis *h*, and the minor principal stress direction coincides with axis *v*. The rotation of the specimen between two consecutive stages of deformation changed the orientation of the principal stress axes relative to the material on axes *h* and *v*. Therefore, even though there was no rotation of the principal stress axes relative to the material during each stage of deformation, these axes were not fixed in the material throughout the entire multistage process of deformation. Along with a high level of strain, it is a feature of SPD processes. The objective of this paper is to reveal a possible effect of this change in the direction of the principal stresses relative to the material on the evolution of material properties.

The V-shape dies were made of heat-treated X210Cr12 steel. The essential geometric parameters of each die are shown in Figure 4. No lubricant between the dies and specimens was used in this study. Both Rastegaev’s method and SPD process were conducted at room temperature on a Sack and Kiesselbach hydraulic press with a capacity of 6.3 MN.

### 3.2. Specimens

The specimens for the SPD process are made of low carbon steel (Thyssenkrupp Materials doo, Indjija, Serbia) (C15E, according to EN standards [33]) whose chemical composition obtained by an optical emission spectrometer ARL 2460 (method OES OES SRPS C. A1.011 (2004), Thermo Fisher Scientific, Waltham, MA, USA) is shown in Table 1.

The cross-section of the initial specimens is a square whose side was 14 mm. The initial length of the specimens was 70 mm. These dimensions suggest that it is reasonable to assume that the state of strain is similar to that of plane strain. The specimens were machined from a rod of diameter 28 mm, heat-treated at 910 °C for 30 min, and cooled down in the furnace for 24 h. After cooling, a square-shaped 7 × 7 mesh (mesh cell is 2 × 2 mm) was applied on the billet face for measuring the components of the strain tensor.

The specimens for Rastegaev’s method were made of the same material. The initial diameter and height of the cylinders were 20 mm.

### 3.3. Measurements

The press stroke and forming load were recorded by a data acquisition system Spider 8 Hottinger Baldwin Messtechnik.

The microstructure was examined by scanning electron microscope (SEM) (JEOL, JSM-6460LV, Tokyo, Japan) and transmission electron microscope (TEM) (Thermo Fisher Scientific, Tecnai G2 200 kV, Waltham, MA, USA). For SEM, the cross-sections were prepared according to the standard metallographic procedure. The samples were cut from upset specimens using a Discotom machine (Struers, Discotom, Bellerup, Denmark) and ground (Knuth Rotor, Struers) with Struers silicon carbide grinding paper (up to 2000 grit). Polishing was carried out on a DP-U2 (Struers) polishing machine in three preliminary steps with MetaDi (Buehler, MetaDi, Lake Bluff, IL, USA) diamond suspension with 6, 3, and 1 µm diamond particles. The final polishing step was carried out with MetaDi Supreme (Buehler) 0.25 µm diamond particles. Before SEM observations, the samples were plated with gold (up to 5 nm thick). SEM was operated with 20 kV accelerating voltage. For TEM, 7 × 5 × 0.2 µm samples were made by a Quanta 3D FEG focused ion beam (FIB). Before the material removal by FIB, a 12 × 1 × 1 µm protective layer was applied on the sample surface. Different electric currents were used for rough and fine material removal ranging from 15 nA to 1 nA. TEM was operated in bright field mode and 200 kV accelerating voltage. Firstly, the microstructure was compared at the intersection of axis *c* (Figure 3) with two cross-sections as follows. Plane *C* generates one of these cross-sections. The other was generated by plane *vh* at a distance of about 3 mm from the free surface. The microstructure at these two locations is illustrated in Figure 5. It is evident from this figure that the difference in the microstructure is small. Therefore, subsequent measurements that are discussed in Section 4 were made in plane *C*.

The Vickers microhardness measurements were performed according to the ISO 6507-1 standard [34] on a Huayin HVS-1000A microhardness tester with a load of 100 gf (0.9807 N, HV 0.1). The measurements were made in plane *C* at several points on the longer axis (*h* or *v* in Figure 3). The distance between the neighboring points for measuring the microhardness was 1 mm. In the vicinity of each point, five microhardness measurements were made to ensure the reliability of results. The distance between indentations is at least 300 µm. The highest and lowest values were discarded, and the arithmetic mean of the three remaining numbers was taken as the Vickers microhardness.

## 4. Results

### 4.1. Strain

The shape of the stress-free face of the specimen and mesh cells after several upsetting stages is presented in Figure 6. The area of cross-sections parallel to plane *C* (Figure 3) slightly reduced with the number of stages. Therefore, the process was not precisely plane strain. The equivalent strain, $\varepsilon_{e}$, is adopted as a measure of strain. In what follows, the equivalent strain is determined in mesh cells located along the lines that are the intersections of the stress-free face with planes *H* and *V* (Figure 3).

Therefore, the increment of the equivalent strain in the course of stage *i* of the process is found from the equation
(1)$$\Delta \varepsilon_{e}=\sqrt{\frac{2}{3}}\sqrt{{\left[\ln\left(\frac{L_{i+1}}{L_{i}}\right)\right]}^{2}+{\left[\ln\left(\frac{W_{i+1}}{W_{i}}\right)\right]}^{2}+{\left[\ln\left(\frac{L_{i+1}}{L_{i}}\right)+\ln\left(\frac{W_{i+1}}{W_{i}}\right)\right]}^{2}}.$$

Here $L_{i}$, $L_{i+1}$, $W_{i}$ and $W_{i+1}$ are determined from the geometry of the cell at the beginning and the end of stage *i* (Figure 7). In deriving Equation (1), the equation of incompressibility is used. The value of the equivalent strain at the end of the process is found by summing $\Delta \varepsilon_{e}$ for all stages.

The highest value of the equivalent strain after the eighteenth stage of the process is at the intersection of the stress-free surface and axis *c*, $\varepsilon_{e}\approx 3.38$ (Figure 8). After the eighth stage, the cross-sectional area of the specimen did not change significantly (Figure 6c,d).

Figure 9 presents the dependence of the equivalent strain increment calculated using Equation (1) at the intersection of the stress-free surface and axis *c* on the stage of the process. The highest value was obtained at the second stage, $\Delta \varepsilon_{e}\approx 0.66$. The value of $\Delta \varepsilon_{e}$ gradually decreased at subsequent stages and this value is negligible after the eighth stage.

### 4.2. Microstructure

The initial microstructure (Figure 10) consisted of ferrite (dark area) and pearlite (ferrite and cementite, light area). Grain size measurements were carried out according to EN ISO 643:2012 standard [35], and it was found that the average ferrite grain size was about 19 µm. The volume fraction of ferrite and pearlite was 85% and 15%, respectively.

The change in microstructure observed throughout the process of deformation was not uniform over the cross-section generated by plane *C* (Figure 3). The discussion below focuses on two areas. One area is in the vicinity of axis *c*. The other area is in the region of the contact surface between the workpiece and tool.

Slight elongation of ferrite grains and pearlite colonies together with rearranging of pearlite colonies was noticed over the entire cross-section after the second stage (Figure 11a,b). However, slight dimming of some ferrite grain boundaries was identified in the vicinity of axis *c* only (Figure 11a). After the eighth stage, additional elongation of both constituents in the vicinity of axis *c* was observed together with a further dimming of ferrite grain boundaries (Figure 11c,d). The microstructure after the eighth, twelfth, and eighteenth stages became more irregular and dimmed. As the number of stages increased, the difference between the microstructure in the areas mentioned above increased, with the ferrite grains and pearlite in the vicinity of axis *c* being more deformed and refined compared to the other area. Some small cracks and voids were also noticed, which is typical for heavily deformed microstructures [36]. Considering the pearlite colonies, the SPD process resulted in elongation and fragmentation of pearlite in the vicinity of axis *c*. The microstructural observations were in good agreement with the distribution of the equivalent strain, which attains a maximum at the axis (see Figure 8).

TEM micrographs and corresponding selected area electron diffraction (SAED) of the ferrite area in the vicinity of axis *c* are present in Figure 12. At the end of the second stage (Figure 12a), the ferrite microstructure mainly consisted of parallel bands of elongated subgrains having a width of 100–200 nm, which is quite similar to the microstructure generated in [37] by ECAP processing. The band boundaries were predominantly in low-angle misorientations. It was also apparent that interior dislocation cells were also present inside the bands. This kind of boundary microstructure is typical in heavily deformed metals. The TEM microstructures and corresponding ring patterns are shown in Figure 12b–d after the eighth, twelfth, and eighteenth upsetting stages. Equiaxed ferrite grains with an average size of 150–400 nm were observed in all three cases. The presence of equiaxed grains, i.e., the ferrite grains with high angle boundaries, was confirmed by ring patterns with a large number of reflections and a TEM contrast at the grain boundaries. However, the grain refinement after the eighteenth stage was not significant compared to the twelfth.

### 4.3. Microhardness

The initial microhardness was 139 HV. The dependence of the microhardness on the process stage is depicted in Figure 13 at four points on the longer axis (*h* or *v* in Figure 3). The corresponding values of the equivalent strain are also shown in Figure 13. The microhardness attained its maximum value of 258 HV at axis *c* (Figure 3) during the sixth stage of the process.

A non-monotonic function describes the distribution of the microhardness in the subsequent stages. On the other hand, the microhardness monotonically decreased as the distance from axis *c* increased.

### 4.4. SPD Process Versus Rastegaev’s Method

The microhardness at axis *c* did not change significantly after the second stage of the process (Figure 13). On the other hand, it is rather difficult to uniformly deform the cylinder using Rastegaev’s method to a value of the equivalent strain much larger than 2. Therefore, the material properties of specimens at the end of the fourth stage of the SPD process were compared to the corresponding properties after Rastegaev’s method. The equivalent strain in Rastegaev’s method is $\varepsilon_{e}=\ln\left(H_{0}/H_{f}\right)$ where $H_{0}$ and $H_{f}$ are the initial and final height of the cylinder, respectively. The equivalent strain at the end of the fourth stage of the SPD process was equal to 2 (Figure 13). Then, since $H_{0}$ was given, the required final height of the cylinder was determined from the equation $H_{f}=H_{0}e^{-2}$.

The microhardness measured on specimens deformed by Rastegaev’s method was 243.2 HV. This value is practically the same as that measured on the specimen after the SPD process (Figure 13).

A comparison of the microstructures generated by the SPD process and Rastegaev’s method is presented in Figure 14. In the case of Rastegaev’s method, ferrite grains were heavily compressed in the direction of the external force (Figure 14b). The microstructure of these grains was practically uniform over the entire volume of the specimen. The pearlite colonies were slightly affected by the higher strains in this test, particularly the cementite within the pearlite.

The microstructure generated after the fourth stage of the SPD process was not visually highly deformed (Figure 14a). The strain was compensated by some microstructural conformation, which resulted in a dimming of grain boundaries (Figure 11).

It is seen from Figure 14 that the two processes considered results in the same evolution of microhardness but different evolutions of microstructure. This means that there is no apparent correlation between the microhardness and microstructure and that an adequate theoretical description of the evolution of microstructure in SPD processes requires constitutive equations that involve a measure of the rotation of principal stress axes relative to the material.

In general, it is a controversial issue whether or not there is any correlation between microhardness and microstructure. In [38], it has been mentioned that the high dislocation density closely correlates with high values of microhardness. The processing of cylindrical tubular parts by rotary extrusion has been studied in [39]. It has been concluded that low-sized grains created in the middle region of the tube are the reason for the higher values of microhardness compared to other regions. In addition, the processing of AISI 1045 steel by helical rolling-pressing has created the material with the smallest grains and the highest microhardness as compared to ECAP and helical rolling [40]. However, the processing of commercial 5483 Al–5Mg alloy by HPT has refined the microstructure, but finer grains have not necessarily resulted in higher microhardness [41]. In particular, coarser grains and higher microhardness have been observed at certain points of the specimen. Similarly, a combination of extrusion and ECAP of AM60 magnesium alloy has resulted in lower microhardness values after the second pass, even though the grain size has monotonically reduced with the number of passes [42].

## 5. Discussion

The primary objective of this research is to compare the evolution of material properties in the SPD process and Rastegaev’s method of cylinder upsetting. The conceptual difference between these two processes is that the principal stress axes rotate relative to the material in the former, and they are fixed in the material in the latter. Taken together, the findings of the present paper suggest that:The rotation of the principal stress axes does not affect the evolution of microhardness (i.e., the value of microhardness at two points is the same if the value of the equivalent strain at these points is the same) in the case of the two processes studied.The rotation of the principal stress axes affects the evolution of microstructure (i.e., parameters that characterize the microstructure at two points are different even if the value of the equivalent strain at these points is the same).

## Figures and Tables

**Figure 1 materials-13-04667-f001:**
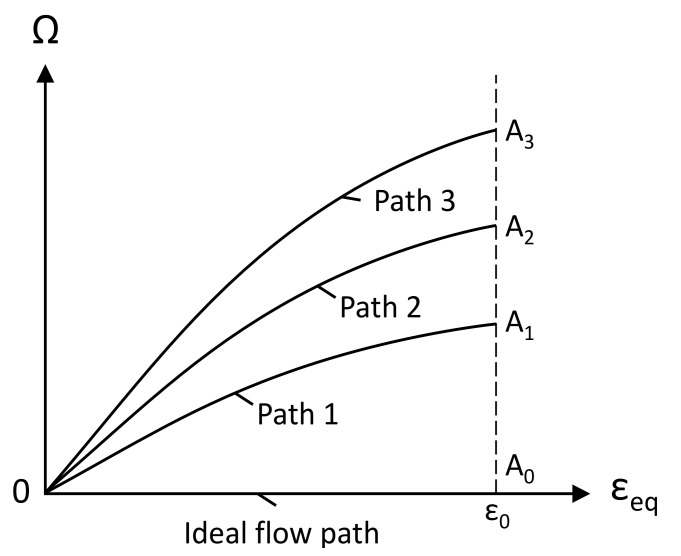
Several paths that result in the same equivalent strain at the end of the process.

**Figure 2 materials-13-04667-f002:**
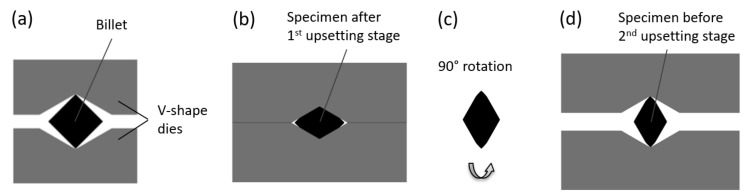
Schematic diagram of the upsetting by V-shape dies: (**a**) billet between the dies (**b**) first stage of the process (**c**) rotation of the specimen (**d**) second stage of the process.

**Figure 3 materials-13-04667-f003:**
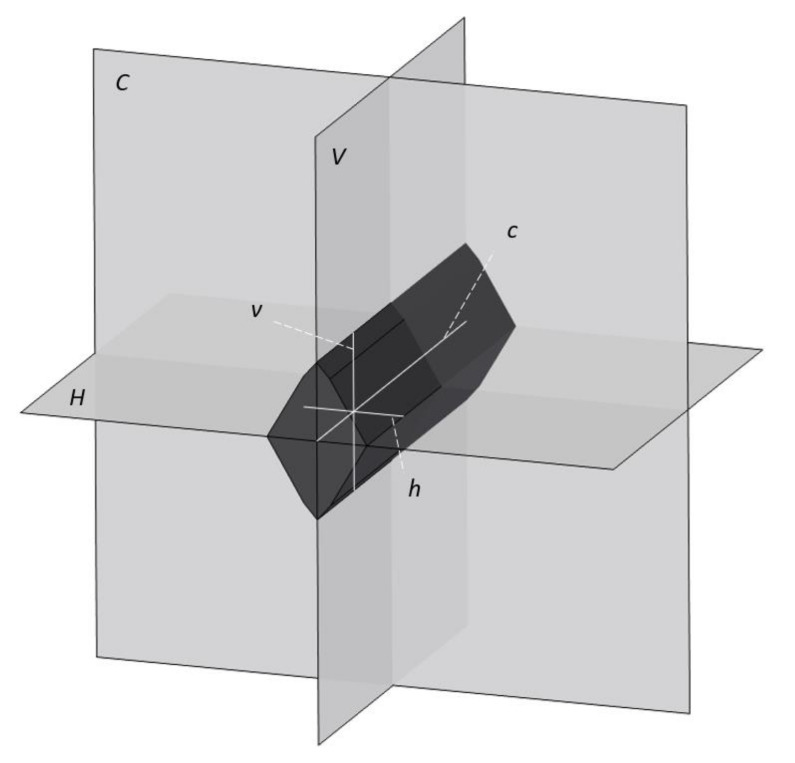
Planes and axes that are further used for describing the locations where measurements are done.

**Figure 4 materials-13-04667-f004:**
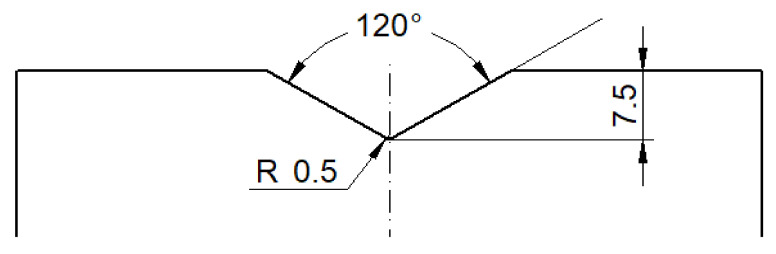
V-shape die geometry (the dimensions are in mm).

**Figure 5 materials-13-04667-f005:**
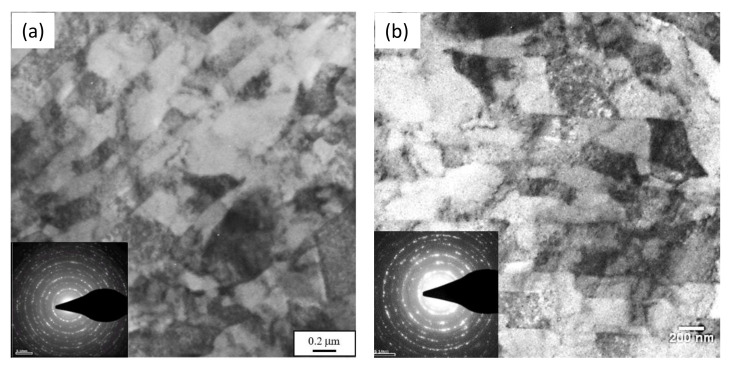
Comparison of the microstructure at two locations: (**a**) plane *C*, (**b**) near the free surface.

**Figure 6 materials-13-04667-f006:**
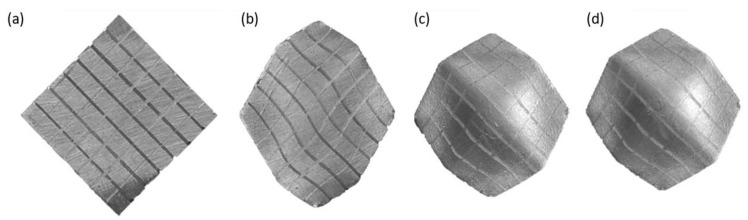
Cross-section and mesh cells (**a**) initial specimen, (**b**) after the second stage, (**c**) after the eighth stage, (**d**) after the twelfth stage.

**Figure 7 materials-13-04667-f007:**
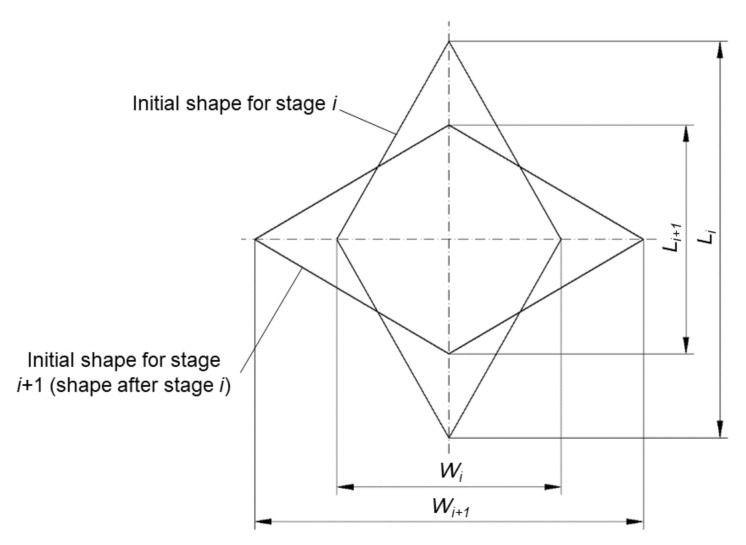
Geometry of mesh cells in two subsequent stages of the process.

**Figure 8 materials-13-04667-f008:**
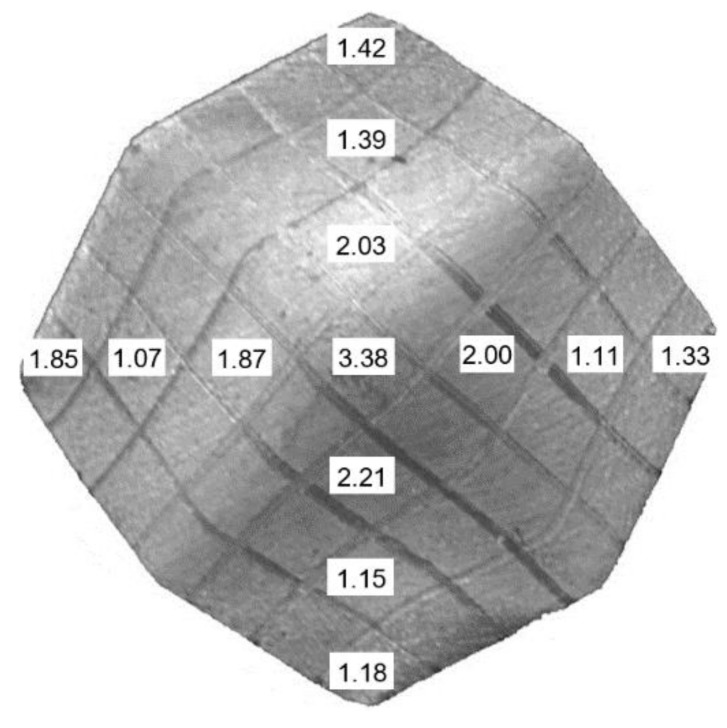
Distribution of the equivalent strain after the eighteenth stage of the process.

**Figure 9 materials-13-04667-f009:**
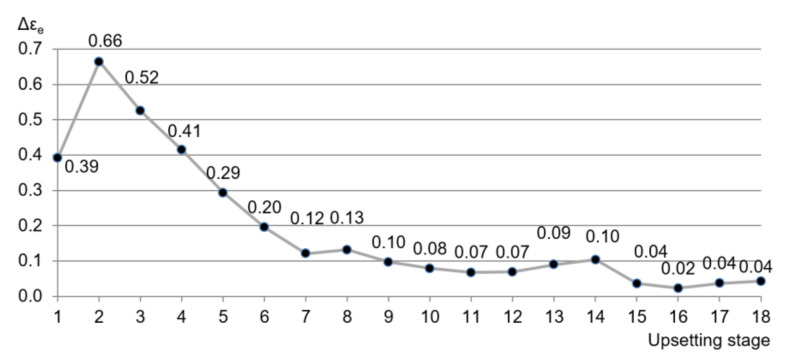
Dependence of $\Delta \varepsilon_{e}$ on the stage of the process at axis *c*.

**Figure 10 materials-13-04667-f010:**
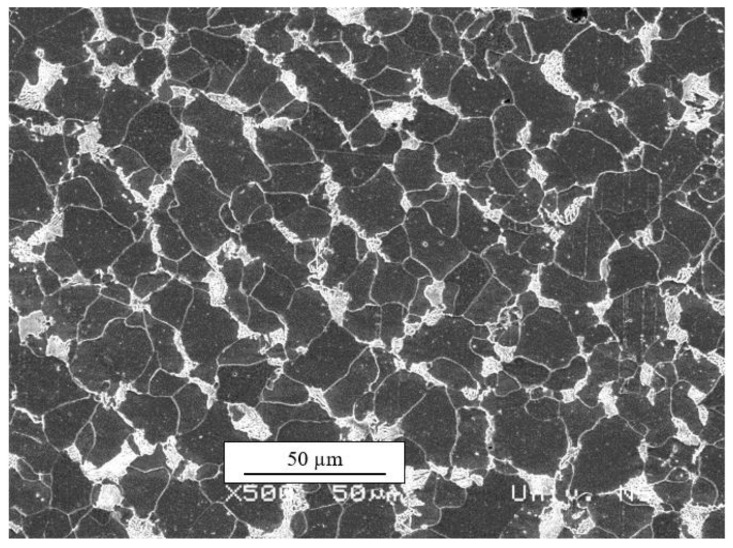
Initial microstructure (SEM, 500×).

**Figure 11 materials-13-04667-f011:**
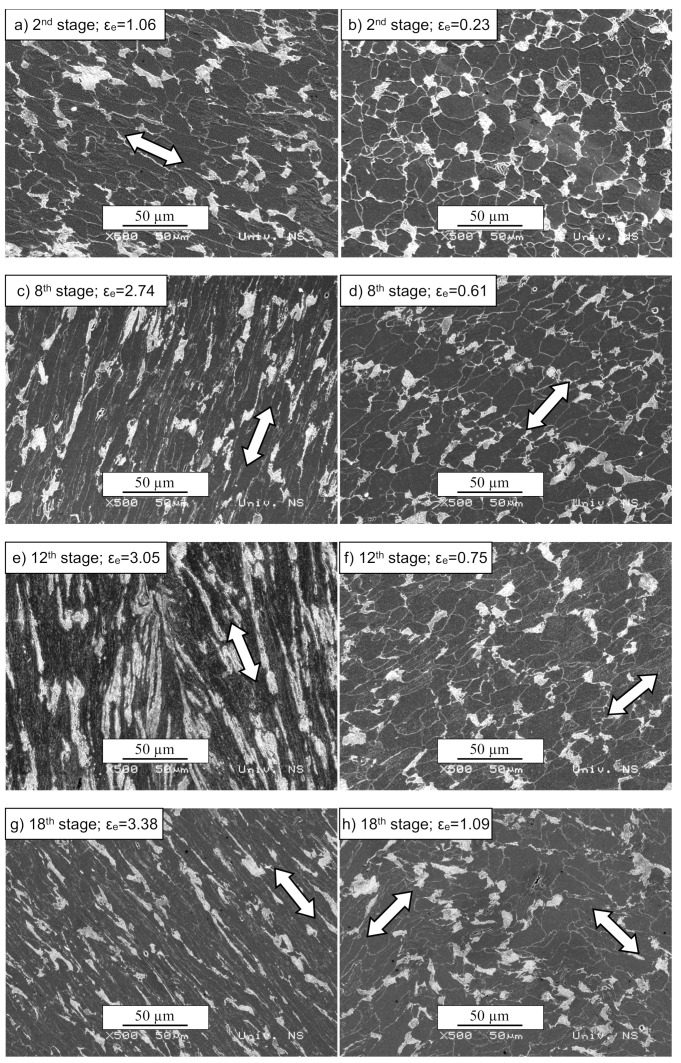
SEM microstructure after several stages of the process (500×). The images on the left correspond to axis *c*, and on the right to the contact surface vicinity. (**a**) 2^nd^ stage, ε_e_ = 1.06, (**b**) 2^nd^ stage, ε_e_ = 0.23, (**c**) 8^th^ stage, ε_e_ = 2.74, (**d**) 8^th^ stage, ε_e_ = 0.61, (**e**) 12^th^ stage, ε_e_ =3.05, (**f**) 12^th^ stage, ε_e_ =0.75, (**g**) 18^th^ stage, ε_e_ =3.38, (**h**) 18^th^ stage, ε_e_ =1.09.

**Figure 12 materials-13-04667-f012:**
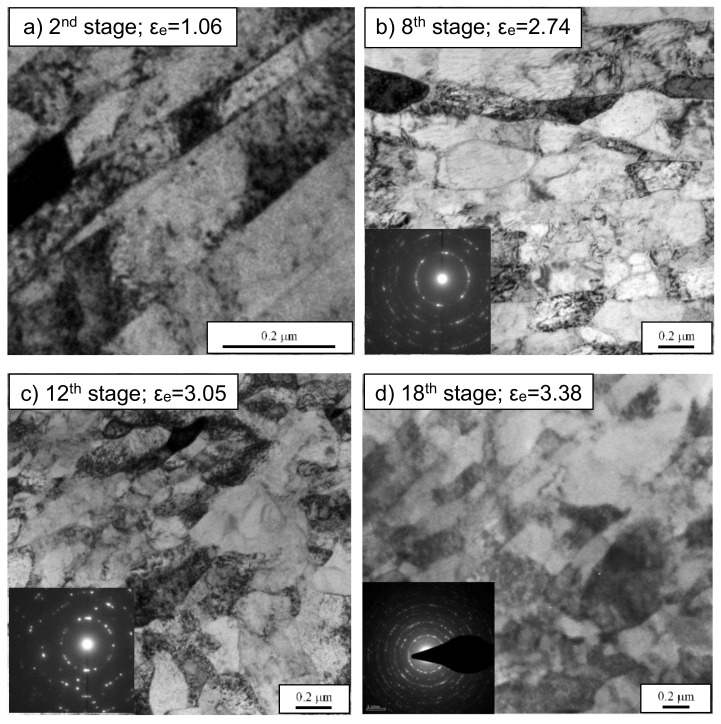
TEM micrograph and selected area electron diffraction (SAED) patterns at axis *c* after several stages of the process. (**a**) 2^nd^ stage, ε_e_ = 1.06, (**b**) 8^th^ stage, ε_e_ = 2.74, (**c**) 12^th^ stage, ε_e_ = 3.05, (**d**) 18^th^ stage, ε_e_ = 3.38.

**Figure 13 materials-13-04667-f013:**
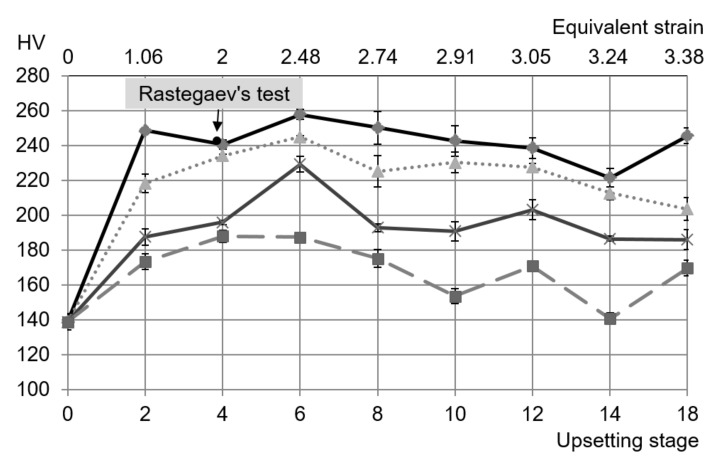
Dependence of microhardness on the process stage at several locations (
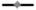

*c*-axis, 
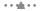
 —2 mm from *c*-axis, 
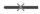
 —4 mm from *c*-axis, 
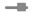
 —6 mm from *c*-axis).

**Figure 14 materials-13-04667-f014:**
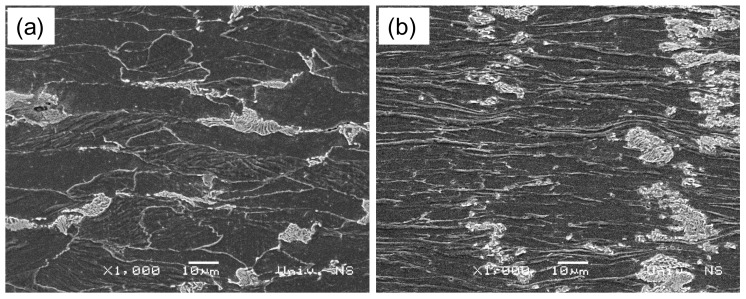
SEM microstructure (1000×): (**a**) severe plastic deformation (SPD) process and (**b**) Rastegaev’s method.

**Table 1 materials-13-04667-t001:** Chemical composition of C15E steel.

Mass. %	C	Si	Mn	S	Cr	P	Al	Cu	Mo	Ni
C15E steel	0.17	0.25	0.516	0.019	0.017	0.015	0.022	0.140	0.045	0.214

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
