# Peer review of "Effect of Rotation of the Principal Stress Axes Relative to the Material on the Evolution of Material Properties in Severe Plastic Deformation Processes"

_materials, 2020, doi:10.3390/ma13204667_

Round 1

Reviewer 1 Report

row 258 "The pearlite colonies are practically not affected by plastic deformation in this test." Strain affected the pearlite, mainly at higher overall strains the cementite within the pearlite deforms plastically, than, you rewrite the sentence. row 264 4. Discussion and Conclusions Conclusions must be comprehensive and not written like a report. Some parts must be rewritten. All part from 265-268 delete. row 276 up to 292, part incorporate to section 3. Results.

Some question: - what is the main aim of the paper? - microstructure and microhardness; findings?

Reviewer 2 Report

1/ Abstract. The information provided in the first part of Abstract should be placed in Introduction. Abstract should be limited to the description of own methods and results. There is no correlation to the original work of the authors. Abstract must be completely updated.

2/ In Introduction you mentioned only typical SPD methods. Note that there are also other more advanced methds, which allow to produce refined microstructure partd of larger size. For example, see: /doi.org/10.2298/JMMB190910008R.

3/  Figure 4 should be placed in Results. Use the same markers in figures a) and b) for magnifications.

4/ There is no methodology provided for microstructure investigations, both for transmission electron microscopy and scanning electron microscopy, grinding, polishing, etching, microscope applied, observation modes applied, parameters, etc …?

5/ Figure 9. For the steel containing 0.17% C, probably dark area is ferrite and light contrast is pearlite ? You have an adverse description in the text.

6/ Figure 11. What is the idenfication of the phases in this figure. Diffraction patterns are not solved; They need more attention and explanation.

7/ Figure 12; Rather strain in this graph should be proportional to better assess as hardness is changing in following deformation stages. Reconsider presenting this picture

8/ Conclusions: „The second conclusion means that there is no apparent correlation between the microhardness and microstructure …” It is not true. It is very typical in different types of materials. You just ignored dislocation processes occuring in the microstructure like dynamic recovery that lead to some softening during severe plastic deformation. Reconsider this because it is crucial for the manuscript. Without this this manuscript is not true in its microscopic part.

Reviewer 3 Report

This is an interesting work however there is lack of details and presentation style. Some comments that may help to improve this work are as follow:

The introduction is very vague and not clear why is there need for this work. I have noted only “The present paper attempts to apply this method experimentally.” I think the rotational effect. It is indicated that the author propose a succinct and critical introduction in which also is highlighted the need of this work.

There was stated as novelty the “effect of SPD on material properties” but this was studied in most of literature works

Any industrial applications ?

“For SEM, the cross-sections are prepared according to the standard metallographic procedure.” OK, but which is this one ? cause otherwise difficult to replicate .. Please provide details

Also the sample preparation for TEM is very vague

Figure 4 is very poor presented and discussed, it is difficult to understand what it shows

“Therefore, the process is not precisely plane strain.” Somehow very confusion phrase, if is not plan strain what is ?

Please describe the method to measure the grain size for Figure 9.

It is better to indicate with arrow grain elongation in Figure 10

Please separate the discussion and conclusions cause they are little bit confusing   

Round 2

Reviewer 1 Report

The author does not follow comments from the reviewer.

Reviewer 2 Report

The authors addressed most of my comments. The current version is improved and the paper can be recommended for publishing in its current form.

Reviewer 3 Report

.